# Risk factors for bovine rotavirus infection and genotyping of bovine rotavirus in diarrheic calves in Bangladesh

Nasir Uddin Ahmed[1], Abul Khair[1,2], Jayedul Hassan[3], Md. Abu Hadi Noor Ali Khan[4], A. K. M. Anisur Rahman [1], Warda Hoque[5], Mustafizur Rahman[5], Nobumichi Kobayashi[6], Michael P. Ward [7], Md. Mahbub Alam [1]*

1 Department of Medicine, Faculty of Veterinary Science, Bangladesh Agricultural University, Mymensingh, Bangladesh, 2 IUBAT-International University of Business Agriculture and Technology, Dhaka, Bangladesh, 3 Department of Microbiology and Hygiene, Faculty of Veterinary Science, Bangladesh Agricultural University, Mymensingh, Bangladesh, 4 Department of Pathology, Faculty of Veterinary Science, Bangladesh Agricultural University, Mymensingh, Bangladesh, 5 Infectious Diseases Division, Virology Laboratory, icddr,b, Mohakhali, Dhaka, Bangladesh, 6 Department of Hygiene, School of Medicine, Sapporo Medical University, Sapporo, Japan, 7 Sydney School of Veterinary Science, The University of Sydney, Camden, New South Wales, Australia

* asamahbub2003@yahoo.com

**Data Availability Statement:** All relevant data are within the paper and its Supporting information files.

## Abstract

Bovine rotavirus (BRV) is considered the leading cause of calf diarrhea worldwide, including Bangladesh. In this study we aimed to identify risk factors for BRV infection and determine the G and P genotypes of BRV strains in diarrheic calves. Fecal samples were collected from 200 diarrheic calves in three districts between January 2014 and October 2015. These samples were screened to detect the presence of BRV using rapid test-strips BIO K 152 (RTSBK). The RTSBK positive samples were further tested by polyacrylamide gel electrophoresis and the silver staining technique to detect rotavirus dsRNA. Risk factors were identified by multivariable logistic regression analysis. The G and P genotypes of BRV were determined by RT-PCR and sequencing. A phylogenetic tree was constructed based on the neighbor-joining method using CLC sequence viewer 8.0. About 23% of the diarrheic calves were BRV positive. The odds of BRV infection were 3.8- (95% confidence interval [95% CI]: 1.0–14.7) and 3.9-times (95% CI:1.1–14.2) higher in Barisal and Madaripur districts, respectively, than Sirrajganj. The risk of BRV infection was 3.1-times (95% CI: 1.5–6.5) higher in calves aged ≤ 5 weeks than those aged >5 weeks. Moreover, the risk of BRV infection was 2.6-times (95% CI:1.1–5.8) higher in crossbred (Holstein Friesian, Shahiwal) than indigenous calves. G6P[11] was the predominant genotype (94.4%), followed by G10P[11] (5.6%). The BRV G6 strains were found to be closest (98.9–99.9%) to Indian strains, and BRV G10 strains showed 99.9% identities with Indian strain. The VP4 gene of all P[11] strains showed >90% identities to each other and also with Indian strains. The most frequently identified BRV genotype was G6P[11]. About 23% of calf diarrhea cases were associated with BRV. To control disease, high-risk areas and younger crossbred calves should be targeted for surveillance and management. The predominant genotype could be utilized

**Funding:** MMA received fund to conduct this study [Grant number is: 2015/17/BAU]. The study was funded by Bangladesh Agricultural University Research System. The funders had no role in study design, data collection and analysis, decision to publish, or preparation of the manuscript.

**Competing interests:** The authors have declared that no competing interests exist.

as the future vaccine candidate or vaccines with the dominant genotype should be used to control BRV diarrhea in Bangladesh.

## Introduction

Rotaviruses are recognized as the major causative agent of severe diarrhea in infants and children, and the young of a variety of mammalian and avian species throughout the world [1, 2]. Bovine rotavirus (BRV) is the leading cause of calf diarrhea worldwide [3]. Among all causal agents of diarrhea, BRV alone is responsible for 62.5% of diarrhea outbreaks in beef and dairy herds [4]. Globally, >30% of all rotavirus-related deaths occurs in India, Bangladesh and Pakistan [5].

The rotavirus (RV) genome consists of 11 segments of double-stranded RNA and encodes six structural (segments 1–4 encode VP1-VP4 proteins, segment 6 encodes VP6 protein, and segment 9 encodes VP7 protein) and six nonstructural (NSP1-NSP6) proteins. Segments 5, 7, 8 and 10 encode nonstructural protein NSP1, NSP3, NSP2 and NSP4, respectively. However, segment 11 encodes NSP5 or NSP 6. These viruses have been classified into 10 genetically distinct groups (A-J) [6]. The electrophoretic migration pattern of the eleven RNA segments in polyacrylamide gel (RNA pattern) is specific to each rotavirus group. Groups A, B, C, and H infect both humans and animals, while Groups D, E, F, and G infect animals and birds [3]. Group A rotaviruses are members of the genus Rotavirus, family Reoviridae [7]. They are classified based on antigenic and genetic differences of the outer capsid antigens, VP7 and VP4 and the inner capsid protein, VP6. Two viral surface proteins, VP4 (a protease-cleaved, or P protein) and VP7 (a glycoprotein, or G protein) are the targets of neutralizing antibodies. These proteins may mediate protection induced by rotavirus vaccines. The antigenicity of group A rotavirus strains has been described by the dual classification system with G-type and P-type. At least 41 G-types and 57 P-types, based on the nucleotide sequences of VP7 and VP4 genes, have been described to date in rotaviruses from humans and various animal species (https://rega.kuleuven.be/cev/viralmetagenomics/virus-classification/rcwg). In human rotaviruses, the major genotypes are G1, G2, G3, G4 and G9, which are combined with P[4], P[6], and P[8] [8]. Although at least six P genotypes—(P6[1], P7[5], P8[11], P11[14], P[17] and P[21]) and 8 G genotypes—(G1, G3, G5, G6, G7, G8, G10, and G15) have been reported among bovine RV-group A [9, 10], only G6, G10 and G8 combined with P[5], P[11], and P[1] are considered epidemiologically important [4]. The most common worldwide BRVA genotypes are considered to be G6 (range, 39.8–78.3%), followed by G10 (21%) in the Americas, Europe, Asia and Australia, and G8 (3%) in Africa. Regarding P type, P[5] strains (range, 37.1–50.0%) are the most prevalent in Europe, the Americas, Asia, and Australia followed by P[11] (range, 15.4–34.8%) and P[1] (2%). A total of 20 individual G and P combinations have been described so far and three combinations, G6P[5], G6P[11] and G10P[11], are predominant (combined prevalence, 40%) in many areas of the world [11]. Surveillance of BRV is very important for disease prevention and more specifically for the development of a vaccine. Continuous monitoring of emerging and re-emerging BRV strains is essential for a better understanding of the viral ecology within a region, and to improve the implemented vaccination programs by updating vaccine strains.

The prevalence of rotavirus infection varies according to risk factors such as herd size, and the timing and amount of colostrum feeding. The prevalence was reported to be relatively higher in small scale farms than medium and large scale farms in Ethiopia [12]. In contrast,

the prevalence of calf scour in general was reported to be higher in larger farms than medium sized farms in Argentina [13]. The feeding of colostrum within one hour of calving has been reported to have a protective effect on rotavirus infection [12]. One study revealed that none of the calves were infected with BRV in herds in which > 2 liters colostrum were provided [12]. There are few reports on the prevalence of bovine rotavirus in Bangladesh [14–17]. Human G and P genotypes have been described in Bangladesh [18, 19], but there is only one report on bovine G and P genotypes [20]. Knowledge of risk factors and circulating bovine G and P genotypes will help in the selection of vaccine candidate strains in the future. Hence the objectives of this study were to identify the risk factors for bovine rotavirus infection, determine the distribution of bovine rotavirus G and P genotypes and understand the genetic diversity of bovine rotaviruses prevalent in diarrheic calves in Bangladesh dairy farms.

## Materials and methods

### Ethics statement

The study protocol was approved by the Animal Welfare and Experimentation Ethical Committee (AWEEC) of Bangladesh Agricultural University (AWEEC/BAU/2013/01). Oral consent was taken from the owner of the calves before sampling.

### Sample and data collection

Fecal samples were collected from 0–3 month old diarrheic calves. A total of 200 samples were collected by convenience sampling from Barishal, Madaripur, and Sirajgonj districts of Bangladesh during 2014–2015. A single sample was collected from each herd. More than two thirds of the herds were from subsistence managment system and the remaining were from large dairy herds. The samples were aseptically collected in sterile plastic bottles and transported to the laboratory in an icebox. Data on age, sex, breed and location of the calves were recorded using a questionnaire during sampling. Fecal samples were preserved at -20˚C until further examination. The age of the calf was determined from the herd record book.

### Detection of BRV by rapid detection kit BIO K 152

Fecal samples were screened for the presence of bovine rotaviruses (BRV) by rapid detection kit BIO K 152 (Bio-X Diagnostics, Belgium) following manufacturer's instructions. This is a chromatographic lateral flow immunoassay coated with monoclonal antibodies specific for rotavirus and colored gold colloidal reagents labeled with monoclonal antibodies specific for rotavirus. Briefly, a spoonful of liquid fecal sample was homogenized carefully to prevent foam formation. Solid feces were initially diluted with a dilution solution and similarly homogenized. A test strip was placed in the homogenized solution, and results were recorded after 10 minutes. The appearance of red color at both the C and T line of the strip was considered a positive result.

### Ribonucleic acid polyacrylamide gel electrophoresis (RNA-PAGE)

Fecal samples were diluted with 10% sterile Phosphate Buffered Saline (PBS, pH 7.2). Supernatants from the suspension were collected by centrifugation at 15,000 rpm for 15 minutes. The supernatant was used for the extraction of viral RNA following the protocol described previously [21]. The extracted RNA was subjected to RNA-PAGE for the detection of 11 segments of rotaviral dsRNA as described previously [21, 22]. RNA was resolved in 10% polyacrylamide gels and stained by silver nitrate [21].

## Identification of risk factors

The data on study area, age, sex, season and disease status were entered into Microsoft Excel 2015 and transferred to R 3.5.0 [23] for statistical analysis. Age was converted to a categorical variable based on median age. Months were converted to three seasons. Calves positive in either RTSBK or PAGE-SS was considered to be BRV infected and used as the outcome variable (positive, negative). District, age, breed, sex and season were used as explanatory variables. The Pearson chi-square test was used to assess the association between BRV infection status and explanatory variables. The R functions "table" and "chisq.test" were used to construct contingency tables and to perform chi-square tests, respectively. Any explanatory variable associated with BRV infection with a p-value of $\leq 0.20$ was included in multivariable logistic regression analysis. Collinearity among explanatory variables was assessed by Cramer's phi-prime statistic (R package "vcd," "assocstats" functions). A pair of variables was considered collinear if Cramer's phi-prime statistic was >0.70 [24]. Stepwise multivariable logistic regression was used to identify risk factors for BRV infection. The final multivariable model was automatically selected based on the lowest Akaike's information criterion value. We used the Hosmer-Lemeshow goodness-of-fit test [25] using the "hoslem.test" function of the R package "ResouceSelection" [26] to assess the overall model fit. Confounding was checked by observing the change in the estimated coefficients of the variables in the final model by adding a non-selected variable to the model. If the inclusion of this non-significant variable led to a change of > 25% of any parameter estimate, that variable was considered to be a confounder and kept in the model [27]. The two-way interactions of all variables remaining in the final model were assessed for significance based on AIC values, rather than significance of individual interaction coefficients [27]. The data used for the identification of risk factors is presented in S1 File.

## RNA extraction for reverse transcription polymerase chain reaction (RT-PCR)

Initially, 5% fecal suspension was prepared in sterile PBS and centrifuged at 12,000 rpm for 1 min. The supernatant was used for RNA extraction using QIAamp® RNA Mini kit (Qiagen, Hilden, Germany) according to the manufacturer's instructions.

**RT-PCR of the bovine rotavirus.**   RT-PCR targeting the genes—VP7 and VP4 were used to detect BRV. Full-length VP7 gene product was indicated by 1062 amplicon base pair (bp) size, while 877 bp product showed the presence of partial length VP4 gene segment. PCR amplification of the whole length VP7 gene (1062 bp) was performed using generic primers Bov9Com5 and Bov9Com3 (Table 1) according to the conditions described previously [28]. The PCR products were further checked for the VP7 subtypes (G6, G8, and G10) by a second-round PCR using Bov9Com5 as forward and either of the primers specific for G6, G8 or G10 as the reverse primers, respectively [29]. P typing was performed through the amplification of partial length VP4 gene (877 bp) using primers Con3-5' end and Con2-3' end (Table 1). Similar to that of G typing, a second PCR with the amplified PCR products was performed to determine VP4 gene subtypes (P1, P5, and P11) using the respective primers (Table 1) as described previously [4]. These PCR products of VP7 and VP4 genes were sequenced to determine G and P genotypes and construct a phylogenetic tree.

**Sequencing for G and P genotyping and phylogenetic analysis.**   Sequences of BRV genes encoding VP7 and VP4 were determined directly with RT-PCR products amplified with Bov9-Com5, Bov9Com3, and Con3-5'end, Con2-3'end, respectively. PCR products were purified using ExoSAP-IT (USB Corp, Cambridge, MA). Nucleotide sequencing was carried out in an automated ABI3500 xL Genetic Analyzer (Applied Biosystem, Foster City, CA) and Big Dye Terminator v3.1 Cycle Sequencing Kit (Applied Biosystem), as per kit protocol. The

**Table 1. Oligonucleotide primers used in RT-PCR assay and sequencing of the PCR products.**

| Primers | Sequences (5'-3') | Location | Product size | Reference |
|---|---|---|---|---|
| Bov9Com5 | GGCTTTAAAAGAGAGAATTTCCGTTTGG | 1–28 | 1062 bp | [29] |
| Bov9Com3 | GGTCACATCATACAACTCTAATCT | 1039–1062 | | |
| G6 | CTAGTTCCTGTGTAGAATC | 499–481 | 500 bp | |
| G8 | CGGTTCCGGATTAGACAC | 273–256 | 274 bp | |
| G10 | TTCAGCCGTTGCGACTTC | 714–697 | 715 bp | |
| **VP4 typing primers for RT-PCR and P typing of BRV:** | | | | |
| Con3-5'end | TGG CTT CGC TCA TTT ATA GAC A | 11–32 | 877bp | [4] |
| Con 2–3'end | ATT TCG GAC CAT TTA TAA CC | 868–887 | | |
| P1 K- P1 | ACC AA C GAA CGC GGG GGT G | 264–282 | 624bp | |
| P5 K- P5 | RCC AGG TGT CRC ATC AGA G | 336–354 | 552bp | |
| pB223- P11 | GGA ACG TAT TCT AAT CCG GTG | 574–594 | 314bp | |

electropherogram files were inspected using Chromas 2.23 (Technelysium Pty Ltd, Unit 406, 8 Cordelia St, South Brisbane QLD 4101, Australia). Genetyx-WIN Version 5.1 (Software Development, Tokyo, Japan) was used to perform pairwise alignment and calculate sequence identity of VP7 and VP4 genes from different strains. All amplicons were further verified using Sanger sequencing. An automated genotyping based on VP7 and VP4 sequences was performed using Rota C v2.0 web-based tool for rotavirus classification (http://rotac.regatools.be). The gene sequences obtained were submitted to GenBank (Table 2) and were also confirmed by BLAST search (http://blast.ncbi.nlm.nih.gov/Blast.cgi) through which G and P types were determined and aligned nucleotide sequences were downloaded from three GenBank database. Sequences showing >90% homology with >90% query coverage were aligned and a neighbor-joining tree was constructed using CLC sequence viewer 8.0 (Qiagen Aarhus, Denmark).

## Results

### Prevalence and risk factors for bovine rotavirus

The prevalence and distribution of rotavirus diarrhea in calves is shown in Table 3. The overall prevalence of bovine rotavirus was 22.5% (45/200) (95% Confidence Interval (CI): 17.0–29.0) based on the RTSBK test. District, season, breed and age of calves were associated with BRV ($P<0.20$; Table 3) and therefore included in multivariable logistic regression modelling.

Variables identified as risk fasctos using multivariable logistic regression analysis are presented in Table 4. The odds of BRV infection were 3.8- (95% CI: 1.0–14.7) and 3.9-times (95% CI:1.1–14.2) higher in Barisal and Madaripur districts, respectively, than Sirrajganj. The risk of BRV infection was 3.1-times (95% CI: 1.5–6.5) higher in calves aged ≤ 5 weeks than those aged >5 weeks. In addition, the risk of BRV infection was 2.6-times (95% CI:1.1–5.8) higher in crossbred than indigenous calves (Table 4).

### Characteristics of the rotavirus genotypes

According to the serial interpretation of RTSBK and RNA-PAGE, 15% (30/200) samples were found positive for BRV. PAGE analysis of the RTSBK positive samples revealed a typical migration pattern of 11 segments of rotavirus-A. All the positive samples showed a long pattern of electrophoreses and classified into a single pattern (Fig 1).

**Table 2. Profiles of bovine rotavirus analyzed genetically and their gene sequences in a study of diarrheic calves in Bangladesh.**

| Sample ID | Location (District) | Age (Days) | Rotavirus test strip (BIO K 152) result | Genotype | Accession numbers | |
|---|---|---|---|---|---|---|
| | | | | | VP7 | VP4 |
| BAU-1 | Barisal | 27 | + | G6P[11] | MH140436 | MH140452 |
| BAU-2 | Barisal | 7 | + | ND | | |
| BAU-3 | Barisal | 7 | + | ND | | |
| BAU-4 | Barisal | 28 | + | ND | | |
| BAU-5 | Barisal | 90 | + | ND | | |
| BAU-6 | Barisal | 60 | + | G6P[11] | MH140442 | MW075227 |
| BAU-7 | Madaripur | 45 | + | ND | | |
| BAU-8 | Madaripur | 45 | + | ND | | |
| BAU-9 | Barisal | 45 | + | ND | | |
| BAU-10 | Sirajganj | 45 | + | G6P[11] | MH140437 | MH140455 |
| BAU-11 | Sirajganj | 60 | + | ND | | |
| BAU-12 | Sirajganj | 60 | + | G6P[11] | MH140451 | MH140456 |
| BAU-13 | Madaripur | 90 | + | ND | | |
| BAU-14 | Barisal | 75 | + | XP[11] | | |
| BAU-15 | Madaripur | 60 | + | XP[11] | | |
| BAU-16 | Madaripur | 15 | + | ND | | |
| BAU-17 | Madaripur | 12 | + | XP[11] | | MH140464 |
| BAU-18 | Madaripur | 90 | + | G6P[11] | MH140438 | MH140457 |
| BAU-19 | Barisal | 30 | + | ND | | |
| BAU-20 | Barisal | 60 | + | ND | | |
| BAU-21 | Madaripur | 4 | + | ND | | |
| BAU-22 | Madaripur | 90 | + | ND | | |
| BAU-23 | Barisal | 90 | + | ND | | |
| BAU-24 | Barisal | 27 | + | G6P[11] | MH140439 | MH140458 |
| BAU-25 | Madaripur | 10 | + | ND | | |
| BAU-26 | Madaripur | 45 | + | ND | | |
| BAU-27 | Madaripur | 27 | + | G6P[11] | MW075223 | MW075224 |
| BAU-28 | Madaripur | 66 | + | G6P[11] | MH140440 | MH140453 |
| BAU-29 | Madaripur | 22 | + | G6P[11] | MH140441 | MH140454 |
| BAU-30 | Barisal | 2 | + | ND | | |
| BAU-31 | Madaripur | 8 | + | G6P[11] | MW075225 | MW075226 |
| BAU-32 | Madaripur | 42 | + | ND | | |
| BAU-33 | Madaripur | 50 | + | G6P[11] | MH140443 | MH140459 |
| BAU-34 | Madaripur | 34 | + | G6P[11] | MH140444 | MW075228 |
| BAU-35 | Barisal | 35 | + | ND | | |
| BAU-36 | Madaripur | 20 | + | G6P[11] | MH140445 | MH140460 |
| BAU-37 | Madaripur | 45 | + | XP[11] | | MH140461 |
| BAU-38 | Madaripur | 60 | + | G6P[11] | MH140446 | MH140466 |
| BAU-39 | Madaripur | 37 | + | G6P[11] | MH140449 | MH140465 |
| BAU-40 | Madaripur | 60 | + | G6P[11] | MH140450 | MH140467 |
| BAU-41 | Madaripur | 20 | + | G6P[11] | MH140447 | MH140462 |
| BAU-42 | Barisal | 60 | + | ND | | |
| BAU-43 | Barisal | 19 | + | ND | | |
| BAU-44 | Barisal | 90 | + | XP[11] | | MH140468 |
| BAU-45 | Barisal | 17 | + | G6P[11] | MH140448 | MH140463 |

ND: Not determined, X: G non-typable.

**Table 3. Contingency tables and Pearson's Chi-square test conducted to evaluate the association between explanatory variables and rotavirus diarrhea in calves in Barisal, Madaripur and Sirajganj districts in Bangladesh, 2014–2015.**

| Variables | Category | Positive/Tested | Prevalence (95% Confidence Interval) | P-value |
|---|---|---|---|---|
| District | | | | 0.04 |
| | Barisal | 18/68 | 26.5 (16.8–38.8) | |
| | Madaripur | 24/92 | 26.1 (17.7–36.5) | |
| | Sirajganj | 3/40 | 7.5 (1.9–21.5) | |
| Age (weeks) | | | | 0.02 |
| | Up to 5 (median) | 31/100 | 31.0 (22.3–41.1) | |
| | > 5 | 14/100 | 14.0 (8.1–22.7) | |
| Sex | | | | 0.39 |
| | Male | 22/111 | 19.8 (13.1–28.7) | |
| | Female | 23/89 | 25.8 (17.4–36.4) | |
| Breed | | | | <0.001 |
| | Indigenous | 30/163 | 18.4 (12.9–25.4) | |
| | Cross (Holstein: 10/25; Shahiwal:5/12) | 15/37 | 40.5 (25.2–57.8) | |
| Season | | | | 0.05 |
| | Summer (March to June) | 17/56 | 30.4 (19.2–44.3) | |
| | Rainy (July to October) | 10/74 | 13.5 (7.1–23.9) | |
| | Winter (November to February) | 18/70 | 25.7 (16.3–37.8) | |
| Overall | | 45/200 | 22.5 (17.0–29.0) | |

RT-PCR was applied to determine the distribution of G and P serotypes of bovine rotaviruses prevalent in Bangladesh (Table 4). In this study, VP7 gene (G type) could be amplified from 18 out of 30 rotavirus positive samples (Fig 2a). Based on the subtype-specific RT-PCR and web-based analysis, among the 18 amplicons evaluated, 17 were BRV G6 (94.4%) and 1 BRV G10 (5.6%). This indicates that G6 is the predominant genotype (Table 2). The VP7 gene from the remainder of the 12 rotavirus positive samples could not be amplified with the primers used in this study, classifying them as untypable G types. The gene encoding for VP4 (P type) could be amplified from 23 of the positive samples (Fig 2b). Type-specific RT-PCR and web-based analysis revealed that all the P genotypes belonged to BRV P[11]. Genotyping analysis of bovine rotavirus (G and P) indicated that G6P[11] was the most prevalent genotype (94.4%) followed by G10P[11] (5.6%).

## Phylogenetic analysis

All the strains examined in this study clustered with the cattle rotavirus isolates reported in Bangladesh and India (Figs 3–5). BRVs from 17 calves exhibited extremely high sequence

**Table 4. Risk factors identified in the final multivariable logistic regression analysis for rotavirus infection in diarrheic calves in Bangladesh.**

| Variables | Category | Estimate | SE | Adjusted Odds ratio (95% Confidence Interval) |
|---|---|---|---|---|
| District | Barisal | 1.34 | 0.68 | 3.8 (1.0–14.7) |
| | Madaripur | 1.35 | 0.66 | 3.9 (1.1–14.2) |
| | Sirajganj | - | - | Reference |
| Age (weeks) | Up to 5 (median age) | 1.14 | 0.37 | 3.1 (1.5–6.5) |
| | > 5 | - | - | Reference |
| Breed | Indigenous | - | - | Reference |
| | Cross | 0.95 | 0.41 | 2.6 (1.1, 5.8) |

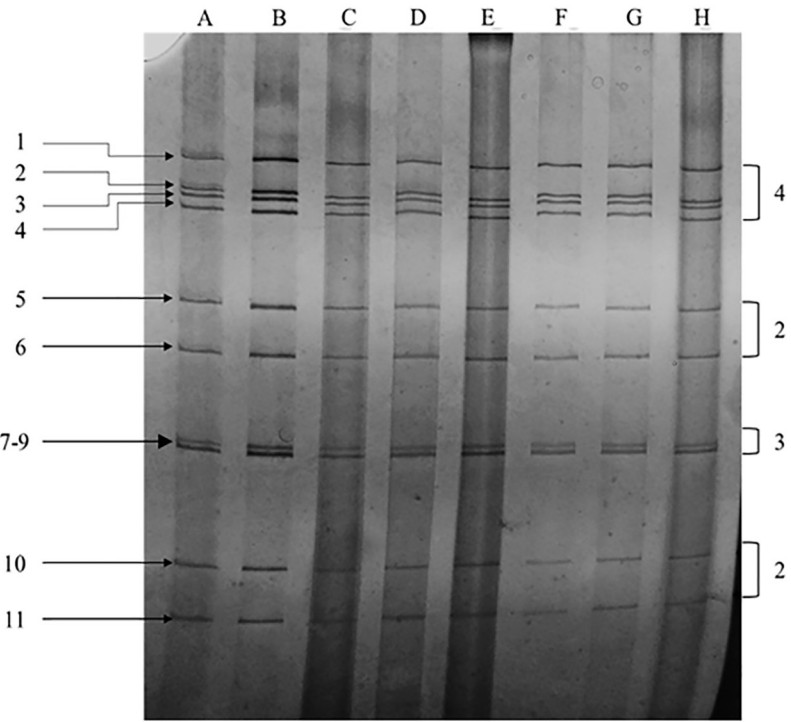

**Fig 1. Electeropherogram of rotavirus strains isolated from diarrhoeic calf samples in Bangladesh.** 1–11, segments of RNA. The genomic RNA segments migration pattern of 4:2:3:2, typical of group A rotavirus was observed in polyacrylamide gel, where segments 7, 8, and 9 moved in a triplet. Segments 1–4, respectively, encode structural protein VP1-VP4, Segemnts 6 and 9 encode structural protein VP6 and VP7, respectively. Segments 5, 7, 8 and 10 encode nonstructural protein NSP1, NSP3, NSP2 and NSP4, respectively. Segment 11 encodes NSP5 or NSP 6. Lane A-H, Rotavirus strains isolated from diarrheic calf samples.

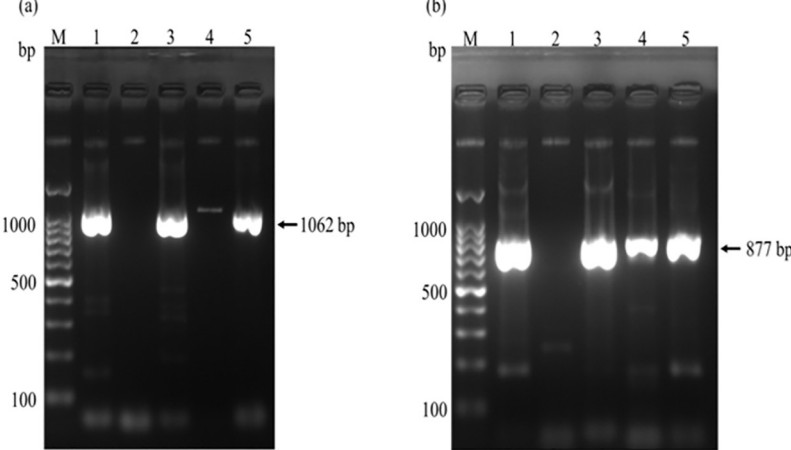

**Fig 2. Representative photographs of RT-PCR amplification of VP7 (a) and VP4 (b) genes.** Lanes: 1–5, suspected diarrheic stool samples. Successful amplification of VP7 and VP4 genes is seen in lane 1, 3, 5, (a) and lane 1, 3, 4, 5 (b), respectively; M, 100 bp DNA ladder (Promega, USA); The product lengths of VP7 and VP4 genes were 1062 and 877 bp, respectively.

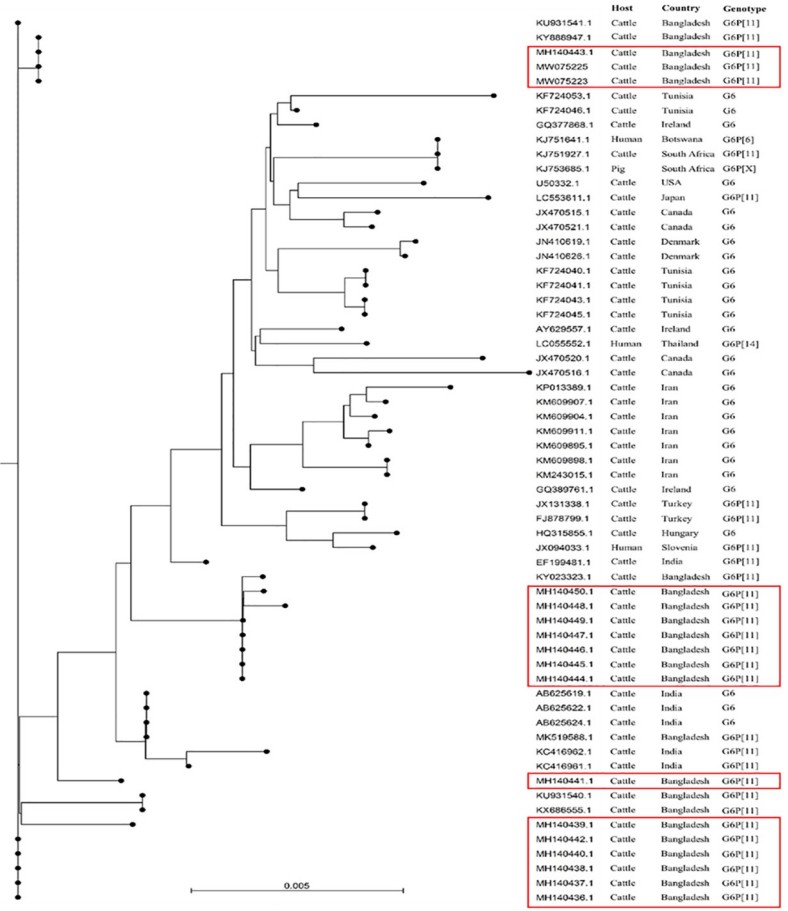

**Fig 3. The evolutionary relationship of G6-VP7 genes of bovine rotaviruses detected in the present study (red rectangle).** The tree was constructed with the aligned sequences downloaded through BLAST search. The relationship was inferred using the Neighbor-Joining method and the ecleotide distances were measure using Kimura 80 model on CLC Sequence viewer 8.0. The scale bar at the bottom indicates nucleotide substitutions per site.

similarities in the VP7 gene (>90%) and were clustered with many strains from Asian countries. VP7 genes of the 17 Bangladeshi BRV G6 strains were the closest to those from India and Iran along with other Asian countries, showing >90% homology (Fig 3).

One G10 BRV from a calf showed >90% identities with Indian strains (Fig 4).

A total of 21 P[11] Bangladeshi strains exhibited >90% VP4 gene sequence identity with each other and also with Indian strains (Fig 5). This result suggests that Bangladeshi BRVs might be of the same origin as those in India.

## Discussion

About 23% of diarrheic calves were found to be positive for BRV. The prevalence of BRV infection varied significantly between regions, breed and age of calves. We have reported the BRV genotypes for the second time in Bangladesh. High-risk areas and crossbred calves ≤ 5 weeks should be targeted for future surveillance and control decisions. The predominant genotype could be utilized as a future vaccine candidate or vaccines with the dominant serotypes should be used to control BRV calf diarrhea in Bangladesh.

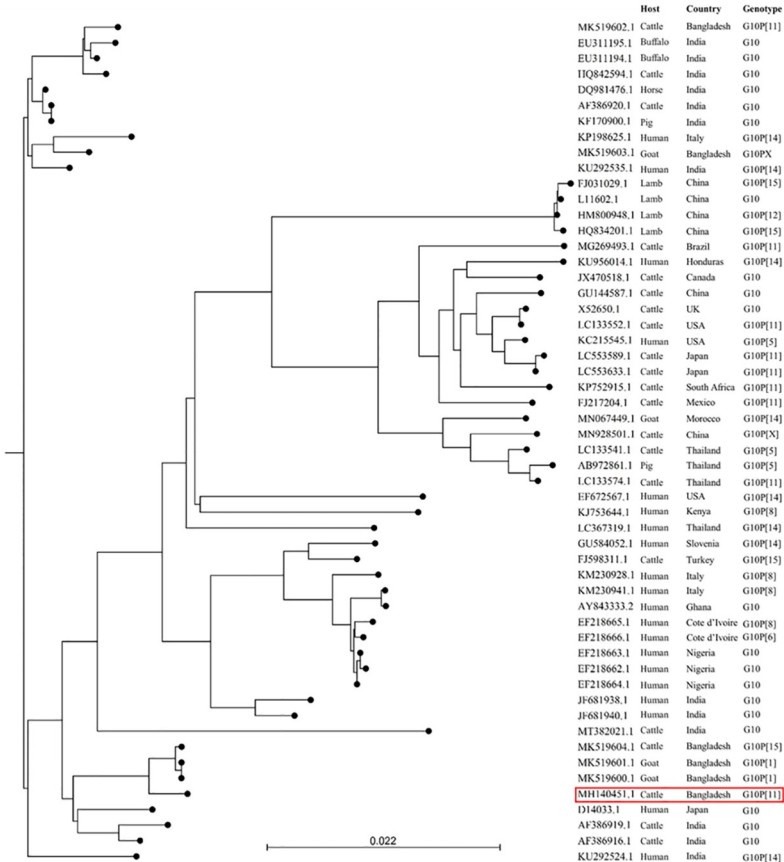

| | Host | Country | Genotype |
|---|---|---|---|
| MK519602.1 | Cattle | Bangladesh | G10P[11] |
| EU311195.1 | Buffalo | India | G10 |
| EU311194.1 | Buffalo | India | G10 |
| IIQ842594.1 | Cattle | India | G10 |
| DQ981476.1 | Horse | India | G10 |
| AF386920.1 | Cattle | India | G10 |
| KF170900.1 | Pig | India | G10 |
| KP198625.1 | Human | Italy | G10P[14] |
| MK519603.1 | Goat | Bangladesh | G10PX |
| KU292535.1 | Human | India | G10P[14] |
| FJ031029.1 | Lamb | China | G10P[15] |
| L11602.1 | Lamb | China | G10 |
| HM800948.1 | Lamb | China | G10P[12] |
| HQ834201.1 | Lamb | China | G10P[15] |
| MG269493.1 | Cattle | Brazil | G10P[11] |
| KU956014.1 | Human | Honduras | G10P[14] |
| JX470518.1 | Cattle | Canada | G10 |
| GU144587.1 | Cattle | China | G10 |
| X52650.1 | Cattle | UK | G10 |
| LC133552.1 | Cattle | USA | G10P[11] |
| KC215545.1 | Human | USA | G10P[5] |
| LC553589.1 | Cattle | Japan | G10P[11] |
| LC553633.1 | Cattle | Japan | G10P[11] |
| KP752915.1 | Cattle | South Africa | G10P[11] |
| FJ217204.1 | Cattle | Mexico | G10P[11] |
| MN067449.1 | Goat | Morocco | G10P[14] |
| MN928501.1 | Cattle | China | G10P[X] |
| LC133541.1 | Cattle | Thailand | G10P[5] |
| AB972861.1 | Pig | Thailand | G10P[5] |
| LC133574.1 | Cattle | Thailand | G10P[11] |
| EF672567.1 | Human | USA | G10P[14] |
| KJ753644.1 | Human | Kenya | G10P[8] |
| LC367319.1 | Human | Thailand | G10P[14] |
| GU584052.1 | Human | Slovenia | G10P[14] |
| FJ598311.1 | Cattle | Turkey | G10P[15] |
| KM230928.1 | Human | Italy | G10P[8] |
| KM230941.1 | Human | Italy | G10P[8] |
| AY843333.2 | Human | Ghana | G10 |
| EF218665.1 | Human | Cote d'Ivoire | G10P[8] |
| EF218666.1 | Human | Cote d'Ivoire | G10P[6] |
| EF218663.1 | Human | Nigeria | G10 |
| EF218662.1 | Human | Nigeria | G10 |
| EF218664.1 | Human | Nigeria | G10 |
| JF681938.1 | Human | India | G10 |
| JF681940.1 | Human | India | G10 |
| MT382021.1 | Cattle | India | G10 |
| MK519604.1 | Cattle | Bangladesh | G10P[15] |
| MK519601.1 | Goat | Bangladesh | G10P[1] |
| MK519600.1 | Goat | Bangladesh | G10P[1] |
| MH140451.1 | Cattle | Bangladesh | G10P[11] |
| D14033.1 | Human | Japan | G10 |
| AF386919.1 | Cattle | India | G10 |
| AF386916.1 | Cattle | India | G10 |
| KU292524.1 | Human | India | G10P[14] |

0.022

**Fig 4. The evolutionary relationship of G10-VP7 genes of bovine rotaviruses detected in the present study (red rectangle).** The tree was constructed with the aligned sequences downloaded through BLAST search. The relationship was inferred using the Neighbor-Joining method and the ecleotide distances were measure using Kimura 80 model on CLC Sequence viewer 8.0. The scale bar at the bottom indicates nucleotide substitutions per site.

## Epidemiology of bovine rotavirus

The overall prevalence of BRV among diarrheic calves (22.5%) is consistent with previous findings of rotaviral infection in various parts of the world. For instance in India, 22% of bovine samples were reported to be positive [30]. Similarly, another study reported 22.7% prevalence in diarrheic calves [29]. In Bangladesh, one study reported 18.3% prevalence of group A rotavirus infections in calves in Dhaka (capital and central) and Mymensingh districts (central north) [16]. In Dhaka district of Bangladesh, the prevalence of rotavirus infection in faeces varied from 7.2% in diarrheic and 6.3% in non-diarrheic calves [14]. Samad and Ahmed [31] reported 12% and 3% prevalence in diarrheic and non-diarrheic calves, respectively in Mymensingh [central-north]. Similarly, 6.2% prevalence of rotaviral infection in diarrheic calves was estimated recently in Netrokona (central north), Dinajpur (north west), and Chattogram (south east) [20]. The overall prevalence of rotavirus in calves, irrespective of diarrheic or non-diarrheic status, was reported to be 5.1% in Chattogram, Cox's Bazar and Rangamati districts (south-east) [32]. Remarkably, the prevalence of this viral infection was found to be much higher in other countries compared to our study, e.g. 63% in Argentina [4], 79.9% in Australia [33], 32.5% in Sudan [34] and 36% in Iraq [35]. Different spatial, temporal and management-related factors might influence the prevalence of BRV infection. The prevalence of

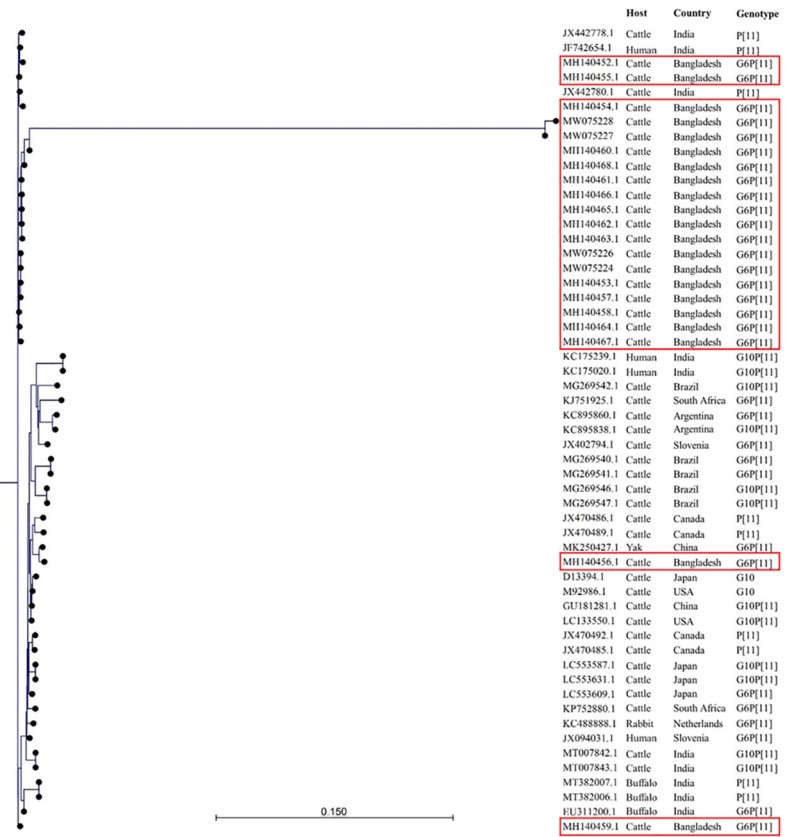

**Fig 5. The evolutionary relationship of P11-VP4 genes of bovine rotaviruses detected in the present study (red rectangle).** The tree was constructed with the aligned sequences downloaded through BLAST search. The relationship was inferred using the Neighbor-Joining method and the nucleotide distances were measure using Kimura 80 model on CLC Sequence viewer 8.0. The scale bar at the bottom indicates nucleotide substitutions per site.

BRV infection might also vary due to differences in study design, sample size, analytical strategy and sensitivity of the diagnostic tests used. In this study, the prevalence of BRV infection was significantly higher in Barishal and Madaripur than Sirajgonj district. This might be due to differences in geographic area, nutritional status of the calves, type of animal rearing, application of hygienic measures in animal sheds and management of the dairy farm.

Calves ≤5 weeks of age are at higher risk for BRV infection compared to those aged > 5 weeks. Similar findings have also been reported by other authors [36, 37]. Age-specific differences in infection are probably due to loss of receptors on enterocytes. The reason for the high occurrence of rotaviral diarrhea under 4–8 weeks of age could be due to a less-developed immune system in neonates and the lack of adequate amounts of maternal antibodies in the colostrum.

Crossbred calves are at higher risk for BRV infection than indigenous calves. The breed of the animal is an important host determinant that influences the immune response and disease severity [27]. Generally, calf diseases are reported to be significantly higher in crossbred than indigenous animals [38]. Although statistically non-significant, the prevalence of BRV infection was 2.4 times higher in crossbred than indigenous calves in Bangladesh [20].

### Genotyping of bovine rotavirus

In this study, 30 bovine positive rotavirus samples were used for genotyping. Out of 30 samples, 18 (60%) were typed as G (VP7) types. Among the G types, 17 (94.4%) and 1 (5.6%) were identified as G6 and G10, respectively. G6 and G10 are the most common G-types reported throughout the world [4, 10, 39]. Many studies have shown that the G6 strain is prevalent in different countries [10, 11, 40, 41]. In contrast to our results, G8 (17.9%) followed by G10 (8%) and G6 (1.6%) were reported as the most prevalent G-genotype of RVA in calves and goats in Bangladesh and India [20, 42]. This difference might be due to the differences in sample sizes and geographic locations of the study.

In this study, 23 (76.7%) of the 30 samples were typed as P (VP4) types and all were P[11]. In contrast, P[1] (11.3%) was reported as the most frequent P-genotype followed by P[11] (3.2%) and P[15] (1.6%) in another study from Bangladesh [20].

Similar to our findings, P[11] was also reported as the most prevalent (93.9%, 31/33) genotype in India [43]. However, P[5] was identified as the predominant (66%) serotypes in one study [10].

### Genetic evolution of bovine rotaviruses

The phylogenetic analysis of the viruses detected in this study indicated that Bangladeshi and Indian isolates are clustered in the same lineage and distantly related with another lineage. G6 (VP7) genotype was highly identical to each other and also with Indian strains. In the case of G10 (VP7), Bangladeshi and Indian strains were identical. For the P[11] genotype, all Bangladeshi and Indian isolates were grouped within the same lineage. Bangladeshi and Indian isolates showed >90% identities at both nucleotide and amino acids levels. In addition, it was observed that all three Bangladeshi isolates have maximum identities of up to 98% at nucleotide and amino acids levels with the Indian isolates. Bangladesh and India share the fifth longest international land border (about 4,096 kilometers, including Assam, Tripura, Mizoram, Meghalaya and West Bengal). The West Bengal–Bangladesh border alone is 2,217 km. Legal and illegal movement of rotavirus-infected animals from India to Bangladesh might be frequent [44]. Therefore, the relatedness of Bangladeshi isolates with the Indian isolates is expected due to transboundary spread of the viruses.

A limitation of this study was that we included only three out of 64 districts which represent only 5% of Bangladesh. We identified G6 and P[11] strains as prevalent genotypes whereas another study determined G8 and P[1] as prevalent genotypes [20]. This indicates that genetic diversity of BRV in calves exists in Bangladesh. Ongoing surveillance of BRV is required to understand the true prevalence and dominant genotypes in Bangladesh.

## Conclusion

The most frequently identified bovine rotavirus genotype in Bangladesh was G6P[11]. About a quarter of the calf diarrhea cases was associated with BRV. High-risk areas and younger crossbred calves should be targeted for future surveillance and control decisions. The predominant genotype could be utilized as a future vaccine candidate or vaccines with the dominant genotype should be used to control BRV calf diarrhea in Bangladesh.

## Supporting information

**S1 File. Data used for the identification of risk factors of bovine rotavirus infection in diarrheic calves.**
(CSV)

**S1 Raw images.**
(PDF)

## Acknowledgments

The authors are grateful to the farmers for providing data and samples from their animals.

## Author Contributions

**Conceptualization:** Md. Mahbub Alam.

**Data curation:** Nasir Uddin Ahmed, Abul Khair.

**Formal analysis:** Nasir Uddin Ahmed, A. K. M. Anisur Rahman, Michael P. Ward.

**Funding acquisition:** Md. Mahbub Alam.

**Investigation:** Nasir Uddin Ahmed, Warda Hoque, Mustafizur Rahman.

**Methodology:** Nasir Uddin Ahmed, Abul Khair, Jayedul Hassan, A. K. M. Anisur Rahman, Nobumichi Kobayashi, Michael P. Ward, Md. Mahbub Alam.

**Project administration:** Md. Mahbub Alam.

**Resources:** Abul Khair, Md. Abu Hadi Noor Ali Khan, Warda Hoque, Mustafizur Rahman, Nobumichi Kobayashi.

**Software:** Jayedul Hassan, A. K. M. Anisur Rahman, Nobumichi Kobayashi.

**Supervision:** Md. Abu Hadi Noor Ali Khan, Md. Mahbub Alam.

**Visualization:** Jayedul Hassan.

**Writing – original draft:** Nasir Uddin Ahmed, Jayedul Hassan.

**Writing – review & editing:** Nasir Uddin Ahmed, Abul Khair, Jayedul Hassan, Md. Abu Hadi Noor Ali Khan, A. K. M. Anisur Rahman, Warda Hoque, Mustafizur Rahman, Nobumichi Kobayashi, Michael P. Ward, Md. Mahbub Alam.

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
