## [Decision Letter · Decision Letter 0]

6 Dec 2021

PONE-D-21-31987Risk factors for bovine rotavirus infection and genotyping of bovine rotavirus in diarrheic calves in BangladeshPLOS ONE

Dear Dr. Alam,

Thank you for submitting your manuscript to PLOS ONE. After careful consideration, we feel that it has merit but does not fully meet PLOS ONE’s publication criteria as it currently stands. Therefore, we invite you to submit a revised version of the manuscript that addresses the points raised during the review process.

We look forward to receiving your revised manuscript.

Kind regards,

Vibhav Gautam

Academic Editor

PLOS ONE

Journal Requirements:

● A clean copy of the edited manuscript (uploaded as the new *manuscript* file).

Additional Editor Comments (if provided):

Dear Dr. Alam,

Please check the attached comments. As you can see that both the reviewers have suggested minor comments for the improvement of your manuscript.

Thanks

Reviewers' comments:

Reviewer's Responses to Questions

**Comments to the Author**

1. Is the manuscript technically sound, and do the data support the conclusions?

Reviewer #1: Partly

Reviewer #2: Yes

2. Has the statistical analysis been performed appropriately and rigorously? 

Reviewer #1: Yes

Reviewer #2: Yes

3. Have the authors made all data underlying the findings in their manuscript fully available?

Reviewer #1: Yes

Reviewer #2: Yes

4. Is the manuscript presented in an intelligible fashion and written in standard English?

Reviewer #1: Yes

Reviewer #2: Yes

5. Review Comments to the Author

Reviewer #1: Minor comments

Abstract

1. The age of the calves should be preferably in weeks over months.

2. Crossbred calves details like types, distribution among types, etc. missing.

3. Comparison to Indian strain of G6 and G10 w.r.t. specific locations in genotype should be elaborated.

Introduction

1. There are lot of lines where space between words are missing. For example, Line 50. "Bovinerotavirus" should be "Bovine rotavirus"; Line 100. "...,breedand location..." should be "..., breed and ..."; Line 108. "..placedin.." should be "...placed in...". Similarly for Line 116. 153. 162. 171. 200. 201. 202. 214. 226. 258. 304.

2. Line 52. RV is not abbreviated before.

3. The flow of introduction should be re-looked.

4. Line 166. Details of Chromas 2.23 required.

5. Line 178. "ND: Note determined" should be "Not Determined"

6. Table 3. Age should be presented in weeks instead of month. Breed details are required. Are Cross Breed unique or represent non-indigenous as a whole?

7. Line 244. "This result...vice-versa" suggests transit of BRVs via established routes. Please revise the sentence appropriately.

8. Line 224. "Blast" should be "BLAST"

9. Line 314. "Legal and illegal...frequent". The author may substantiate this statement with reference.

10. What is the detection limit of the PCR-test used.

11. Different vaccines have been developed and deployed across the world. Serum Institute of India has made bovine rotavirus vaccine as well. The authors may comment upon the utility of known vaccines on the cohort tested.

12. What is the vaccination status of the cohort and that of the herd from which the calves were selected? What criteria was used to determine selection of calves in the herd?

13. How does the data correlate with Indian rotaviral reports like that of Tatte et a. 2019 (PMID: 30879319) and Kumar et al 2018 (PMID: 29464925), Chitambar 2011 (PMID: 20880637)?

14. Have the authors identified any similarities with the human rotaviruses?

Reviewer #2: In the present study, Ahmed et al. has tried to identify G and P genotypes of bovine rotavirus (BRV) strains in diarrheic calves as well as the major risk factors of BRV infection. In this study, Ahmed et al. noticed the predominant role of G6P[11] genotype (94.4%), followed by G10P[11] (5.6 %) in BRV infection. Further, their experimental evidence suggested age and breed of claves as major risk factors for the BRV infection.

Comments:

1. Authors should also discuss the impact of other factors particularly, season and sex in the abstract.

2. In the introduction section, authors have discussed more about the genotypes of BRV; it will be good, if they will reduce this part and include a paragraph on the impact of different parameters (such as Herd size, colostrum feeding, colostrum timing and colostrum amount) on the rate of BRV infection in calves.

3. If possible, authors should also examine the impact of other factors on BRV infection, which are also identified as one of the potential factors (such as Herd size, colostrum feeding, colostrum timing and colostrum amount) for BRV infection.

4. In this study, author found that female calves are more susceptible for BRV infection while studies have also shown that male calves are more susceptible for BRV infection, what could be the possible reason; please discuss this in to the discussion section.

6. Discuss the principle of rapid detection kit BIO K 152 behind the detection of BRV in the materials and methods.

7. Authors should elaborate the methods used in the study for better understanding of the readers.

8. In figure 1, authors have mentioned 11 RNA fragments but in the figure only 10 RNA segments are visible, 8th RNA segment in all groups is not visible. Please justify this. It will be good if authors will mention about the encoded proteins of different RNA fragments in figure as well as in the discussion section.

9. In figure 2, label the name of gene (VP7 and VP4) along with product length.

10. Author have stated two contradictory statements, Page 3, line No 3-4, and Page 12, line No 201-203, please clarify this.

11. The pattern of cited references in the list is not consistent, please do the needful corrections.

12. There are few grammatical and typographical mistakes in the manuscript, so authors should carefully read the manuscript and do the needful corrections.

6. PLOS authors have the option to publish the peer review history of their article (what does this mean?). If published, this will include your full peer review and any attached files.

Reviewer #1: No

Reviewer #2: **Yes: **Dr. Ajay Kumar, Assistant Professor, Department of Zoology, Institute of Science, Banaras Hindu University, Varanasi-221005

---

## [Author Response · Author response to Decision Letter 0]

29 Dec 2021

We thank both reviewers for their valuable comments to improve the manuscript. The changes made in the revised manuscript were shown as blue font.

Reviewer #1: Minor comments 

Abstract

1. The age of the calves should be preferably in weeks over months.

Response: The Age was categorized to weeks as suggested. Lines 36 and Tables 2 and 3.

2. Crossbred calves details like types, distribution among types, etc. missing.

Response: The details of crossbred calves have been given in Table 2.

3. Comparison to Indian strain of G6 and G10 w.r.t. specific locations in genotype should be elaborated.

Response: Elaborated as suggested by the reviewer. Lines 39-40.

Introduction

1. There are lot of lines where space between words are missing. For example, Line 50. "Bovinerotavirus" should be "Bovine rotavirus"; Line 100. "...,breedand location..." should be "..., breed and ..."; Line 108. "..placedin.." should be "...placed in...". Similarly for Line 116. 153. 162. 171. 200. 201. 202. 214. 226. 258. 304.

Response: Sorry for this unintentional mistake. They were corrected in the revised manuscript.

2. Line 52. RV is not abbreviated before.

Response: RV was elaborated at line 55.

3. The flow of introduction should be re-looked.

Response: The flow of the introduction was relooked.

4. Line 166. Details of Chromas 2.23 required.

Response: The details of the Chromas 2.23 were added in line 184.

5. Line 178. "ND: Note determined" should be "Not Determined"

Response: Corrected as Not Determined.

6. Table 3. Age should be presented in weeks instead of month. Breed details are required. Are Cross Breed unique or represent non-indigenous as a whole?

Response: Age was presented as weeks instead of months in the Table 3. The composition of cross breed calves was also provided in the Table 3.

7. Line 244. "This result...vice-versa" suggests transit of BRVs via established routes. Please revise the sentence appropriately.

Response: This sentence was revised as suggested by the reviewer. Lines: 267-268.

8. Line 224. "Blast" should be "BLAST"

Response: Blast was replaced by BLAST.

9. Line 314. "Legal and illegal...frequent". The author may substantiate this statement with reference.

Response: A reference was added to support this statement. Reference number 44.

10. What is the detection limit of the PCR-test used?

Response: We used conventional PCR so the detection limit is not applicable.

11. Different vaccines have been developed and deployed across the world. Serum Institute of India has made bovine rotavirus vaccine as well. The authors may comment upon the utility of known vaccines on the cohort tested.

Response: Serum Institute, India developed reasserted bovine pentavalent vaccine by using human G1, G2, G3, G4, and G9 serotypes in the genetic background of bovine rotavirus. This vaccine is intended for human use only. Although the name of the vaccine is bovine rotavirus pentavalent vaccine but principally this is not for animal use. We have mentioned in the submitted manuscript that the available vaccines having dominant genotypes in Bangladesh can also be used for our cattle population.

12. What is the vaccination status of the cohort and that of the herd from which the calves were selected? What criteria was used to determine selection of calves in the herd?

Response: Vaccination against rotavirus calf diarrhea has not yet initiated in Bangladesh. The calves were selected from the herd by a convenience sampling technique as we mentioned in the matherials and methods section. Line: 105.

13. How does the data correlate with Indian rotaviral reports like that of Tatte et a. 2019 (PMID: 30879319) and Kumar et al 2018 (PMID: 29464925), Chitambar 2011 (PMID: 20880637)?

Response: Tatte et al., reported 27.3% G10, while we detected only 3.3% G10. In addition the authors also reported P[6] and P[8] while we detected only p[11]. Similarly, Chitambar et al., 2011 reported G8P[14] which we did not detect. . Kumar et al., 2018 reported an unusual bovine rotavirus strain having similarly with human rotavirus which is not similar to our results. We have added this information in the discussion section. 

14. Have the authors identified any similarities with the human rotaviruses?

Response: No, the bovine rotavirus genotypes we reported in this manuscript were not similar with human rotaviruses.

Reviewer #2: In the present study, Ahmed et al. has tried to identify G and P genotypes of bovine rotavirus (BRV) strains in diarrheic calves as well as the major risk factors of BRV infection. In this study, Ahmed et al. noticed the predominant role of G6P[11] genotype (94.4%), followed by G10P[11] (5.6 %) in BRV infection. Further, their experimental evidence suggested age and breed of claves as major risk factors for the BRV infection.

Comments:

1. Authors should also discuss the impact of other factors particularly, season and sex in the abstract.

Response: Season and sex were not significantly associated with BRV infection. So we did not add them in the abstract.

2. In the introduction section, authors have discussed more about the genotypes of BRV; it will be good, if they will reduce this part and include a paragraph on the impact of different parameters (such as Herd size, colostrum feeding, colostrum timing and colostrum amount) on the rate of BRV infection in calves.

Response: We have reduced the introduction section in relation to the BRV genotypes and added one paragraph about the effect of herd size, colostrum feeding, timing and amount. Lines: 87-94.

3. If possible, authors should also examine the impact of other factors on BRV infection, which are also identified as one of the potential factors (such as Herd size, colostrum feeding, colostrum timing and colostrum amount) for BRV infection.

Response: Thank you very much for this comment. It could be valuable if these variables were included in the analysis. However, we don’t have data for these variables.

4. In this study, author found that female calves are more susceptible for BRV infection while studies have also shown that male calves are more susceptible for BRV infection, what could be the possible reason; please discuss this in to the discussion section.

Response: As mentioned in the response to a previous comment of the reviewer that sex was not statistically associated with BRV infection. So we did not describe it in the manuscript.

6. Discuss the principle of rapid detection kit BIO K 152 behind the detection of BRV in the materials and methods.

Response: The principle of rapid detection kit BIO K 152 was added in the materials and method section as suggested by the reviewer. Lines: 113-116.

7. Authors should elaborate the methods used in the study for better understanding of the readers.

Response: We have elaborated risk factor identification methods. Lines: 144-151.

8. In figure 1, authors have mentioned 11 RNA fragments but in the figure only 10 RNA segments are visible, 8th RNA segment in all groups is not visible. Please justify this. It will be good if authors will mention about the encoded proteins of different RNA fragments in figure as well as in the discussion section.

Response: In figure 1, segments 8 and 9 are fused together while segment 7 is separate. So, it appears that there are 10 segments. The gene segments encoded proteins were mentioned in Figure 1 and also in the introduction.

9. In figure 2, label the name of gene (VP7 and VP4) along with product length.

Response: In figure 2, we have added the name of the genes with their product length. This has also been described in lines 161-162 and 166-167,

10. Author have stated two contradictory statements, Page 3, line No 3-4, and Page 12, line No 201-203, please clarify this.

Response: We did not find any contradictory statement in the mentioned places. However, we have rearranged lines 3-4 in page 3.

11. The pattern of cited references in the list is not consistent, please do the needful corrections.

Response: We have corrected the reference lists.

12. There are few grammatical and typographical mistakes in the manuscript, so authors should carefully read the manuscript and do the needful corrections.

Response: We have carefully read the revised manuscript and corrected the typographical mistakes. The grammatical mistakes were corrected by a native English speaking co-author.

---

## [Decision Letter · Decision Letter 1]

26 Jan 2022

PONE-D-21-31987R1Risk factors for bovine rotavirus infection and genotyping of bovine rotavirus in diarrheic calves in BangladeshPLOS ONE

Dear Dr. Alam,

Thank you for submitting your manuscript to PLOS ONE. After careful consideration, we feel that it has merit but does not fully meet PLOS ONE’s publication criteria as it currently stands. Therefore, we invite you to submit a revised version of the manuscript that addresses the points raised during the review process.

We look forward to receiving your revised manuscript.

Kind regards,

Vibhav Gautam

Academic Editor

PLOS ONE

Journal Requirements:

Additional Editor Comments (if provided):

Dear Dr. Alam:

I have received the reports from our advisors on your manuscript, "Risk factors for bovine rotavirus infection and genotyping of bovine rotavirus in diarrheic calves in Bangladesh.

Based on the advice received, I have decided that your manuscript could be reconsidered for publication should you be prepared to incorporate major revisions. When preparing your revised manuscript, you are asked to carefully consider the reviewer comments which are provided below, and submit a list of responses to the comments.

I look forward to receiving your revised manuscript.

Sincerely yours,

Vibhav Gautam

Reviewers' comments:

Reviewer's Responses to Questions

**Comments to the Author**

1. If the authors have adequately addressed your comments raised in a previous round of review and you feel that this manuscript is now acceptable for publication, you may indicate that here to bypass the “Comments to the Author” section, enter your conflict of interest statement in the “Confidential to Editor” section, and submit your "Accept" recommendation.

Reviewer #1: (No Response)

Reviewer #2: (No Response)

Reviewer #3: (No Response)

2. Is the manuscript technically sound, and do the data support the conclusions?

Reviewer #1: Yes

Reviewer #2: Partly

Reviewer #3: Yes

3. Has the statistical analysis been performed appropriately and rigorously? 

Reviewer #1: Yes

Reviewer #2: Yes

Reviewer #3: N/A

4. Have the authors made all data underlying the findings in their manuscript fully available?

Reviewer #1: Yes

Reviewer #2: Yes

Reviewer #3: Yes

5. Is the manuscript presented in an intelligible fashion and written in standard English?

Reviewer #1: No

Reviewer #2: Yes

Reviewer #3: Yes

6. Review Comments to the Author

Reviewer #1: All previous comments have been answered except for minor language corrections and need of citation at one place. Please refer to details here.

line 128 - centrification - should be centrifugation.

line 163 - Rephrase to "RT-PCR targeting the genes - VP7 and VP4 were used to detect BRV."

lines 216-7 - age is represented as months here while in previous sentences ages is mentioned in weeks. Please rectify.

line 232 - Correct "segemnts" to "segments".

Line 238-41 - Rephrase to "Based on the subtype-specific RT-PCR and we-based analysis, among the 18 amplicons evaluated, 17 were BRV G6 (94.4%) and 1 BRV G10 (5.6%). This indicates that G6 is the predominant genotype."

Line 345 - What percentage of the total land of the nation, do the 3 districts cover?

Line 346 - Give reference of the "another study".

Reviewer #2: Authors have not performed the suggested experiments and have also not properly modified the manuscript as per the suggestion. More so, they have not well responded to the comments.

Reviewer #3: Comments:

1) In the present study authors focused only on three districts, what was the status of diarrheic calves in other districts of Bangladesh?

2) As authors said they used BRV rapid detection kit_which only provides the information about the presence and absence of infection, what was the percentage of infection does any other method or technique to calculate the % of infection?

3) In the present study authors find the G and P genotypes in diarrheic calves, if authors prefer the HRM (high resolution melting) analysis in control and infected calves to find out the SNP (single nucleotide polymorphism) which is responsible for the diarrheic condtion??

4) In figure 2a lane no. four there is light amplified product , what represent this band??

Minor comments:

Line no.116,153,171,200,201,258,294 and 304 two-three words are combined ,this might be due to mac or windows please once check it.

7. PLOS authors have the option to publish the peer review history of their article (what does this mean?). If published, this will include your full peer review and any attached files.

Reviewer #1: No

Reviewer #2: No

Reviewer #3: No

---

## [Author Response · Author response to Decision Letter 1]

6 Feb 2022

We thank all reviewers for their valuable comments to improve the manuscript further. The changes we made in the revised version were shown in blue font. 

Reviewer #1: All previous comments have been answered except for minor language corrections and need of citation at one place. Please refer to details here.

line 128 - centrification - should be centrifugation.

Response: centrification was changed to centrifugation in line 128.

line 163 - Rephrase to "RT-PCR targeting the genes - VP7 and VP4 were used to detect BRV."

Response: Rephrased as suggested by the reviewer.

lines 216-7 - age is represented as months here while in previous sentences ages is mentioned in weeks. Please rectify.

Response: Age was converted to weeks according to the suggestion of a reviewer. However, we forgot to replace this in lines 216-217. Now, we have replaced months by weeks in lines 216-217.

line 232 - Correct "segemnts" to "segments".

Response: “Segemnts” was corrected to “segments” in line 232.

Line 238-41 - Rephrase to "Based on the subtype-specific RT-PCR and web-based analysis, among the 18 amplicons evaluated, 17 were BRV G6 (94.4%) and 1 BRV G10 (5.6%). This indicates that G6 is the predominant genotype."

Response: Rephrased as suggested by the reviewer in lines 238-241..

Line 345 - What percentage of the total land of the nation, do the 3 districts cover?

Response: It represents only 5% of Bangladesh. We have added this in lines 353-353..

Line 346 - Give reference of the "another study".

Response: We have added one reference in line 354.

Reviewer #2: Authors have not performed the suggested experiments and have also not properly modified the manuscript as per the suggestion. More so, they have not well responded to the comments.

Response: In fact, it is not clear what the reviewer means by “not performed the suggested experiments and have also not properly modified the manuscript as per the suggestion. The reviewer also mentioned that we did not respond properly to his comments”. As the reviewer did not make any specific comment, we are not able to address to the specific point in the revised manuscript. However, we have attached our response again to every comment of the reviewer in the first revision.

Reviewer comments and response were in the first revision:

1. Authors should also discuss the impact of other factors particularly, season and sex in the abstract.

Response: Season and sex were not significantly associated with BRV infection. So we did not add them in the abstract.

2. In the introduction section, authors have discussed more about the genotypes of BRV; it will be good, if they will reduce this part and include a paragraph on the impact of different parameters (such as Herd size, colostrum feeding, colostrum timing and colostrum amount) on the rate of BRV infection in calves.

Response: We have reduced the introduction section in relation to the BRV genotypes and added one paragraph about the effect of herd size, colostrum feeding, timing and amount. Lines: 87-94.

3. If possible, authors should also examine the impact of other factors on BRV infection, which are also identified as one of the potential factors (such as Herd size, colostrum feeding, colostrum timing and colostrum amount) for BRV infection.

Response: Thank you very much for this comment. It could be valuable if these variables were included in the analysis. However, we don’t have data for these variables.

4. In this study, author found that female calves are more susceptible for BRV infection while studies have also shown that male calves are more susceptible for BRV infection, what could be the possible reason; please discuss this in to the discussion section.

Response: As mentioned in the response to a previous comment of the reviewer that sex was not statistically associated with BRV infection. So we did not describe it in the manuscript.

6. Discuss the principle of rapid detection kit BIO K 152 behind the detection of BRV in the materials and methods.

Response: The principle of rapid detection kit BIO K 152 was added in the materials and method section as suggested by the reviewer. Lines: 113-116.

7. Authors should elaborate the methods used in the study for better understanding of the readers.

Response: We have elaborated risk factor identification methods. Lines: 144-151.

8. In figure 1, authors have mentioned 11 RNA fragments but in the figure only 10 RNA segments are visible, 8th RNA segment in all groups is not visible. Please justify this. It will be good if authors will mention about the encoded proteins of different RNA fragments in figure as well as in the discussion section.

Response: In figure 1, segments 8 and 9 are fused together while segment 7 is separate. So, it appears that there are 10 segments. The gene segments encoded proteins were mentioned in Figure 1 and also in the introduction.

9. In figure 2, label the name of gene (VP7 and VP4) along with product length.

Response: In figure 2, we have added the name of the genes with their product length. This has also been described in lines 161-162 and 166-167,

10. Author have stated two contradictory statements, Page 3, line No 3-4, and Page 12, line No 201-203, please clarify this.

Response: We did not find any contradictory statement in the mentioned places. However, we have rearranged lines 3-4 in page 3.

11. The pattern of cited references in the list is not consistent, please do the needful corrections.

Response: We have corrected the reference lists.

12. There are few grammatical and typographical mistakes in the manuscript, so authors should carefully read the manuscript and do the needful corrections.

Response: We have carefully read the revised manuscript and corrected the typographical mistakes. The grammatical mistakes were corrected by a native English speaking co-author.

Reviewer #3: Comments:

1) In the present study authors focused only on three districts, what was the status of diarrheic calves in other districts of Bangladesh?

The status of BRV infection in diarrheic calves in other districts of Bangladesh was discussed in the discussion section of the first revision. We have highlighted those in lines: 293-301.

2) As authors said they used BRV rapid detection kit which only provides the information about the presence and absence of infection, what was the percentage of infection does any other method or technique to calculate the % of infection?

Response: Initially, we screened all samples by BRV rapid detection kit. The rapid test positive samples were further confirmed by PAGE. So, the percentage of infection according to the serial interpretation of both tests will be 15% (30/200). We have now added this result in the result section. Lines: 223-224.

3) In the present study authors find the G and P genotypes in diarrheic calves, if authors prefer the HRM (high resolution melting) analysis in control and infected calves to find out the SNP (single nucleotide polymorphism) which is responsible for the diarrheic condition?

Response: In our study, all the fecal samples were collected from diarrheic calves, showing symptoms of diarrhea. Sample was not taken from healthy cows. Therefore, comparison of Rotavirus genes between diarrheic cows and healthy cows was not possible, and SNP in VP7 and VP4 was not revealed in our study design.

4) In figure 2a lane no. four there is light amplified product, what represent this band??

Response: We consider that this band was nonspecific amplicon, because the size was a little larger than expected and faint. In the revised manuscript, additional description was inserted to legend of Fig.2 as follows (underlined). “……Lanes: 1-5, suspected diarrheic stool samples. Successful amplification of VP7 and VP4 genes is seen in lane 1, 3, 5 (a) and lane 1, 3, 4, 5 (b), respectively.”

Minor comments:

Line no.116,153,171,200,201,258,294 and 304 two-three words are combined ,this might be due to mac or windows please once check it.

Response: Sorry for this unintentional mistake. We have checked the manuscript and corrected all combined words.

---

## [Editor Report · Decision Letter 2]

14 Feb 2022

Risk factors for bovine rotavirus infection and genotyping of bovine rotavirus in diarrheic calves in Bangladesh

PONE-D-21-31987R2

Dear Dr. Alam,

We’re pleased to inform you that your manuscript has been judged scientifically suitable for publication and will be formally accepted for publication once it meets all outstanding technical requirements.

Kind regards,

Vibhav Gautam

Academic Editor

PLOS ONE
---

## [Editor Report · Acceptance letter]

16 Feb 2022

PONE-D-21-31987R2 

Risk factors for bovine rotavirus infection and genotyping of bovine rotavirus in diarrheic calves in Bangladesh 

Dear Dr. Alam:

I'm pleased to inform you that your manuscript has been deemed suitable for publication in PLOS ONE. Congratulations! Your manuscript is now with our production department. 

Kind regards, 

on behalf of

Dr. Vibhav Gautam 

Academic Editor

PLOS ONE